# Learning Activation Functions for Sparse Neural Networks

Mohammad Loni[*1]  Aditya Mohan[*2]  Mehdi Asadi[3]  Marius Lindauer[2]

[1]Division of Computer Science and Software Engineering, Mälardalen University, Sweden
[2]Institute of Artificial Intelligence, Leibniz University Hannover, Germany
[3]Department of Electrical Engineering, Tarbiat Modares University, Tehran, Iran

**Abstract**  Sparse Neural Networks (SNNs) can potentially demonstrate similar performance to their dense counterparts while saving significant energy and memory at inference. However, the accuracy drop incurred by SNNs, especially at high pruning ratios, can be an issue in critical deployment conditions. While recent works mitigate this issue through sophisticated pruning techniques, we shift our focus to an overlooked factor: hyperparameters and activation functions. Our analyses have shown that the accuracy drop can additionally be attributed to (i) Using ReLU as the default choice for activation functions unanimously, and (ii) Fine-tuning SNNs with the same hyperparameters as dense counterparts. Thus, we focus on learning a novel way to tune activation functions for sparse networks and combining these with a separate hyperparameter optimization (HPO) regime for sparse networks. By conducting experiments on popular DNN models (LeNet-5, VGG-16, ResNet-18, and EfficientNet-B0) trained on MNIST, CIFAR-10, and ImageNet-16 datasets, we show that the novel combination of these two approaches, dubbed S̲parse A̲ctivation F̲unction S̲earch, short: SAFS, results in up to 15.53%, 8.88%, and 6.33% absolute improvement in the accuracy for LeNet-5, VGG-16, and ResNet-18 over the default training protocols, especially at high pruning ratios.[1]

## 1 Introduction

Deep Neural Networks, while having demonstrated strong performance on a variety of tasks, are computationally expensive to train and deploy. When combined with concerns about privacy, energy efficiency, and the lack of stable connectivity, this led to an increased interest in deploying DNNs on resource-constrained devices like micro-controllers and FPGAs (Chen and Ran, 2019).

Recent works have tried to address this problem by reducing the enormous memory footprint and power consumption of DNNs. These include quantization (Zhou et al., 2017), knowledge distillation (Hinton et al., 2015), low-rank decomposition (Jaderberg et al., 2014), and network sparsification using unstructured pruning (a.k.a. Sparse Neural Networks) (Han et al., 2015). Among these, Sparse Neural Networks (SNNs) have shown considerable benefit through their ability to remove redundant weights (Hoefler et al., 2021). However, they suffer from accuracy drop, especially at high pruning ratios; e.g., Mousavi et al. (2022) report ≈54% reduction in top-1 accuracy for MobileNet-v2 (Sandler et al., 2018) trained on ImageNet as compared to non-pruned. While significant blame for this accuracy drop goes to sparsification itself, we identified two underexplored, pertinent factors that can additionally impact it: (i) The activation functions of the sparse counterparts are never optimized, with the Rectified Linear Unit (ReLU) (Nair and Hinton, 2010) being the default choice. (ii) The training hyperparameters of the sparse neural networks are usually kept the same as their dense counterparts.

A natural step, thus, is to understand how the activation functions impact the learning process for SNNs. Previously, Jaiswal et al. (2022) and Tessera et al. (2021) have demonstrated that ReLU reduces the trainability of SNNs since sudden changes in gradients around zero result in blocking

---

[1]Our code is available at `github.com/automl/SAFS`

gradient flow. Additionally, Apicella et al. (2021) have shown that a ubiquitous activation function cannot prevent typical learning problems such as vanishing gradients. While the field of Automated Machine Learning (AutoML) (Hutter et al., 2019) has previously explored optimizing activation functions of dense DNNs (Ramachandran et al., 2018; Loni et al., 2020; Bingham et al., 2020), most of these approaches require a huge amount of computing resources (up to 2000 GPU hours (Bingham et al., 2020)), resulting in a lack of interest in activation function optimization for various deep learning problems. On the other hand, attempts to improve the accuracy of SNNs either use sparse architecture search (Fedorov et al., 2019; Mousavi et al., 2022) or sparse training regimes (Srinivas et al., 2017). To our knowledge, there is no efficient approach for optimizing activation functions on SNN training.

**Paper Contributions**: (i) We analyze the impact of activation functions and training hyperparameters on the performance of sparse CNN architectures. (ii) We propose a novel AutoML approach, dubbed SAFS, to tweak the activation functions and training hyperparameters of sparse neural networks to deviate from the training protocols of their dense counterparts. (iii) We demonstrate significant performance gains when applying SAFS with unstructured magnitude pruning to LeNet-5 on the MNIST (LeCun et al., 1998) dataset, VGG-16 and ResNet-18 networks trained on the CIFAR-10 (Krizhevsky et al., 2014) dataset, and ResNet-18 and EfficientNet-B0 networks trained on the ImageNet-16 (Chrabaszcz et al., 2017) dataset, when compared against the default training protocols, especially at high levels of sparsity.

## 2 Related Work

To the best of our knowledge, SAFS is the first automated framework that tweaks the activation functions of sparse neural networks using a multi-stage optimization method. Our study also sheds light on the fact that tweaking the hyperparameters plays a crucial role in the accuracy of sparse neural networks. Improving the accuracy of sparse neural networks has been extensively researched in the past. Prior studies are mainly categorized as (i) recommending various criteria for selecting insignificant weights, (ii) pruning at initialization or training, and (iii) optimizing other aspects of sparse networks apart from pruning criteria. In this section, we discuss these methods and compare them with SAFS, and briefly review state-of-the-art research on optimizing activation functions of dense networks.

### 2.1 Sparse Neural Network Optimization

**Pruning Insignificant Weights**. A number of studies have proposed to prune the weight parameters below a fixed threshold, regardless of the training objective (Han et al., 2015; Li et al., 2016; Zhou et al., 2019). Recently, Azarian et al. (2020) and Kusupati et al. (2020) suggested layer-wise trainable thresholds for determining the optimal value for each layer.

**Pruning at Initialization or Training**. These methods aim to start sparse instead of first pretraining a dense network and then pruning it. To determine which weights should remain active at initialization, they use criteria such as using the connection sensitivity (Lee et al., 2018) and conservation of synaptic saliency (Tanaka et al., 2020). On the other hand, Mostafa and Wang (2019); Mocanu et al. (2018); Evci et al. (2020) proposed to leverage information gathered during the training process to dynamically update the sparsity pattern of kernels.

**Miscellaneous Sparse Network Optimization**. Evci et al. (2019) investigated the loss landscape of sparse neural networks and Frankle et al. (2020) addressed how it is impacted by the noise of Stochastic Gradient Descent (SGD). Finally, Lee et al. (2020) studied the effect of weight initialization on the performance of sparse networks. While our work also aims to improve the performance of sparse networks and enable them to achieve the same performance as their dense counterparts, we instead focus on the impact of optimizing activation functions and hyperparameters of the sparse neural networks in a joint HPO setting.

## 2.2 Activation Function Search

Inappropriate selection of activation functions results in information loss during forward propagation and the vanishing and/or exploding gradient problems during backpropagation (Hayou et al., 2019). To find the optimal activation functions, several studies automatically tuned activation functions for dense DNNs, being based on either evolutionary computation (Bingham et al., 2020; Basirat and Roth, 2021; Nazari et al., 2019), reinforcement learning (Ramachandran et al., 2018), or gradient descent for devising parametric functions (Tavakoli et al., 2021; Zamora et al., 2022).

Despite the success of these methods, automated tuning of activation functions for dense networks is unreliable for the sparse context since the search spaces for activation functions for dense networks are not optimal for sparse networks (Dubowski, 2020). The same operations that are successful in dense networks can drastically diminish network gradient flow in sparse networks (Tessera et al., 2021). Additionally, existing methods suffer from significant search costs; e.g., Bingham et al. (2020) required 1000 GPU hours per run on NVIDIA® GTX 1080Ti. Jin et al. (2016) showed the superiority of SReLU over ReLU when training sparse networks as it improves the network's gradient flow. However, SReLU requires learning four additional parameters per neuron. In the case of deploying networks with millions of hidden units, this can easily lead to considerable computational and memory overhead at inference time. SAFS, on the other hand, unifies local search on a meta-level with gradient descent to create a two-tier optimization strategy and obtains superior performance with faster search convergence compared to the state-of-the-art.

## 3 Preliminaries

In this section, we develop notations for the later sections by formally introducing the two problems that we address: Network Sparsification and Hyperparameter Optimization.

### 3.1 Network Sparsification

Network sparsification is an effective technique to improve the efficiency of DNNs for applications with limited computational resources. Zhan and Cao (2019) reported that network sparsification could facilitate saving ResNet-18 inference time trained on ImageNet on mobile devices by up to $29.5\times$. Network sparsification generally consists of three stages:

1. *Pre-training*: Train a large, over-parameterized model. Given a loss metric $\mathcal{L}_{train}$ and network parameters $\boldsymbol{\theta}$, this can be formulated as the task of finding the parameters $\boldsymbol{\theta}^{\star}_{pre}$ that minimize $\mathcal{L}_{train}$ on training data $\mathcal{D}_{train}$:

$$\boldsymbol{\theta}^{\star}_{pre} \in \underset{\boldsymbol{\theta} \in \Theta}{\text{argmin}} \left[ \mathcal{L}_{train}(\boldsymbol{\theta}; \mathcal{D}_{train}) \right] \tag{1}$$

2. *Pruning*: Having trained the dense model, the next step is to remove the low-importance weight tensors of the pre-trained network. This can be done layer-wise, channel-wise, and network-wide. The usual mechanisms either simply set a certain percentage of weights (*pruning ratio*) to zero, or learn a Boolean mask $\boldsymbol{m}^{\star}$ over the weight vector. Both of these notions can be generally captured in a manner similar to the dense training formulation but with a separate loss metric $\mathcal{L}_{prune}$. The objective here is to obtain a pruning mask $\boldsymbol{m}^{\star}$, where $\odot$ represents the masking operation and $N$ represents the size of the mask:

$$\boldsymbol{m}^{\star} \in \underset{\boldsymbol{m} \in \{0,1\}^N}{\text{argmin}} \left[ \mathcal{L}_{prune}(\boldsymbol{\theta}^{\star}_{pre} \odot \boldsymbol{m}; \mathcal{D}_{train}) \right] \quad \text{s.t.} \quad \|\boldsymbol{m}^{\star}\|_0 \leq \epsilon \tag{2}$$

where $\epsilon$ is a threshold on the minimal number of masked weights.

3. *Fine-tuning*: The final step is to retrain the pruned network to regain its original accuracy using a fine-tuning [2] loss $\mathcal{L}_{fine}$, which can either be the same as the training loss or a different kind:

$$\boldsymbol{\theta}^{\star}_{fine} \in \underset{\boldsymbol{\theta} \in \Theta}{\mathrm{argmin}} \left[ \mathcal{L}_{fine}(\boldsymbol{\theta}; \boldsymbol{\theta}^{\star}_{pre} \odot \boldsymbol{m}^{\star}, \mathcal{D}_{train}) \right] \tag{3}$$

For the pruning stage, SAFS uses the popular magnitude pruning method (Han et al., 2015) by removing a certain percentage of weights that have a lower magnitude. Compared to structured pruning methods (Liu et al., 2018), the magnitude pruning method provides higher flexibility and a better compression rate $\left( \frac{|\boldsymbol{\theta}^{\star}_{fine}|}{|\boldsymbol{\theta}^{\star}_{pre}|} \times 100 \right)$. Crucially, SAFS is independent of the pruning algorithm; thus, it can optimize any sparse network.

## 3.2 Hyperparameter Optimization (HPO)

We denote the hyperparameter space of the model as $\Lambda$ out of which we sample a hyperparameter configuration $\boldsymbol{\lambda} = (\lambda_1, \ldots, \lambda_d)$ to be tuned by some HPO methods. We assume $c : \boldsymbol{\lambda} \rightarrow \mathbb{R}$ to be a black-box cost function that maps the selected configuration $\boldsymbol{\lambda}$ to a performance metric, such as model-error[3]. HPO's goal can then be summarized as the task of finding an optimal configuration $\boldsymbol{\lambda}^{\star}$ minimizing $c$. Given the fine-tuned parameters $\boldsymbol{\theta}^{\star}_{fine}$ obtained in Equation (3), we define the cost as minimizing a loss $\mathcal{L}_{hp}$ on validation dataset $\mathcal{D}_{val}$ as a bi-level optimization problem:

$$\boldsymbol{\lambda}^{\star} \in \underset{\boldsymbol{\lambda} \in \Lambda}{\mathrm{argmin}} \, c(\boldsymbol{\lambda}) = \underset{\boldsymbol{\lambda} \in \Lambda}{\mathrm{argmin}} \left[ \mathcal{L}_{hp}(\boldsymbol{\theta}^{\star}_{fine}(\boldsymbol{\lambda}); \mathcal{D}_{val}) \right] \tag{4}$$

$$\text{s.t.}$$

$$\boldsymbol{\theta}^{\star}_{fine}(\boldsymbol{\lambda}) \in \underset{\boldsymbol{\theta} \in \Theta}{\mathrm{argmin}} \left[ \mathcal{L}_{fine}(\boldsymbol{\theta}; \boldsymbol{\theta}^{\star}_{pre} \odot \boldsymbol{m}^{\star}, \mathcal{D}_{train}, \boldsymbol{\lambda}) \right]$$

We note that in principle HPO could also be applied to the training of the original model (Equation (1)), but we assume that the original is given and we care only about sparsification.

## 4 Finding Activation Functions for Sparse Networks

The aim of SAFS is to find an optimal hyperparameter configuration for pruned networks with a focus on activation functions. Given the HPO setup described in Section 3.2, we now explain how to formulate the activation function search problem and what is needed to solve it.

### 4.1 Modelling Activation Functions

Using optimization techniques requires creating a search space containing promising candidate activation functions. Extremely constrained search spaces might not contain novel activation functions (*expressivity*) while searching in excessively large search spaces can be difficult (*size*) (Ramachandran et al., 2018). Thus, striking a balance between the expressivity and size of the search space is an important challenge in designing search spaces.

To tackle this issue, we model parametric activation functions as a combination of a unary operator $f$ and two learnable scaling factors $\alpha, \beta$. Thus, given an input $x$ and output $y$, the activation function can be formulated as $y = \alpha f(\beta x)$, which can alternatively be represented as a computation graph shown in Figure 3a.

Figure 1 illustrates an example of tweaking the $\alpha$ and $\beta$ learnable parameters of the *Swish* activation function. We can intuitively see that modifying the suggested learnable parameters for a

---

[2]We use the term fine-tuning interchangeably with re-training
[3]For reasonably sized datasets and models, we estimate this error using k-fold cross-validation.

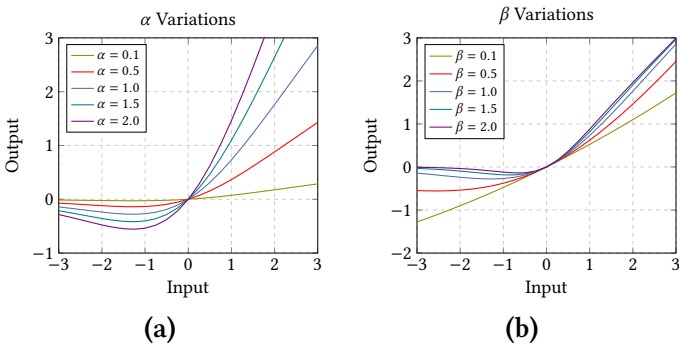

Figure 1: Modifying (a) $\alpha$ and (b) $\beta$ learnable scaling factors of the *Swish* activation function.

sample unary operator provides the sparse network additional flexibility to fine-tune activation functions (Godfrey, 2019; Bingham and Miikkulainen, 2022). Examples of activation functions that we consider in this work have been listed in Appendix E.

For sparse networks, this representation allows efficient implementation as well as effective parameterization. As we explain further in Section 4.2, by treating this as a two-stage optimization process, where the search for $f$ is a discrete optimization problem and the search for $\alpha, \beta$ is interleaved with fine-tuning, we are able to make the search process efficient while capturing the essence of input-output scaling and functional transformations prevalent with activation functions. Note that SAFS falls under the category of adaptive activation functions due to introducing trainable parameters (Dubey et al., 2022). These parameters allow the activation functions to smoothly adjust the model with the dataset complexity (Zamora et al., 2022). In contrast to popular adaptive activation functions such as PReLU and Swish, SAFS automates activation function tuning across a diverse family of activation functions for each layer of the network with optimized hyperparameters.

## 4.2 Optimization Procedure

SAFS performs the optimization layer-wise i.e. we intend to find the activation functions for each layer. Given layer indices $i = 1, \ldots, L$ of the network of depth $L$ an optimization algorithm needs to be able to select a unary operator $f_i^\star$ and find appropriate scaling factors $(\alpha_i^\star, \beta_i^\star)$. We formulate these as two independent objective functions, solved in a two-stage optimization procedure combining discrete and stochastic optimization. Figure 2 shows an overview of the SAFS pipeline.

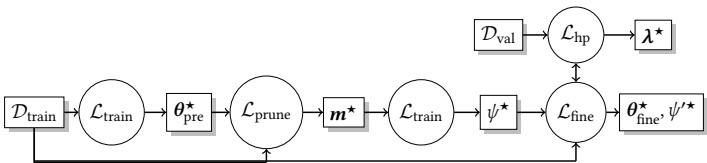

Figure 2: Overview of the entire SAFS pipeline.

**Stage 1: Unary Operator Search.** The first stage is to find the unary operators after the network has been pruned. Crucially, the fine-tuning step happens only after this optimization for the activation function has been completed. We model the task of finding optimal unary operators for each layer as a discrete optimization problem. Given a pre-defined set of functions $F = \{f_1, f_2, \ldots, f_n\}$, we define a space $\mathcal{F}$ of possible sequences of operators $\psi = \langle f_i \mid f_i \in F \rangle_{i \in \{1,\ldots,L\}} \in \mathcal{F}$ of size $L$. Our task is to find a sequence $\psi$ after the pruning stage (Item 2). Since the pre-trained network parameters $\theta_{pre}^\star$ and the pruning mask $m^\star$ have already been discovered, we keep them fixed and use them as an initialization point for activation function optimization. The task is formulated as finding the

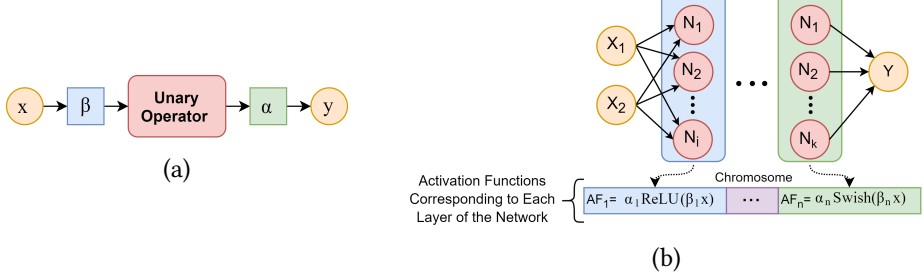

(a)

(b)

Figure 3: (a) SAFS unary activation graph. (b) An example of a solution representing activation functions of each layer in the network.

optimal operators given the network parameters, as shown in Equation (5). During this step, $\alpha$ and $\beta$ parameters are set to 1 to focus on the function class first.

$$\psi^\star \in \underset{\psi \in \mathcal{F}}{\arg\min} \left[ \mathcal{L}_{train}(\boldsymbol{\theta}^\star_{pre} \odot \boldsymbol{m}^\star, \psi; \mathcal{D}_{train}) \right] \tag{5}$$

Given the discrete nature of Equation (5), we use Late Acceptance Hill Climbing (LAHC) (Burke and Bykov, 2017) to iteratively solve it (Please refer to Appendix A for comparison against other search algorithms). LAHC is a Hill Climbing algorithm that uses a record of the history - *History Length* - of objective values of previously encountered solutions in order to decide whether to accept a new solution. It provides us with two benefits: (i) Being a semi-local search method, LAHC works on discrete spaces and quickly searches the space to find unary operators. (ii) LAHC extends the vanilla hill-climbing algorithm (Selman and Gomes, 2006) by allowing worse solutions in the hope of finding a better solution in the future. We represent the design space of LAHC using a chromosome that is a list of activation functions corresponding to each layer of the network. Figure 3b shows an example of a solution in the design space. The benefit of this representation is its flexibility and simplicity. For generating a new search candidate (*mutation operation*), we first swap two randomly selected genes from the chromosome, and then, we randomly changed one gene from the chromosome with a new candidate from the list.

Appendix E lists unary operators considered in this study. To avoid instability during training, we ignored periodic operators (e.g., $cos(x)$) and operators containing horizontal ($y = 0$) or vertical ($x = 0$) asymptotes (e.g., $y = \frac{1}{x}$).

The process of selecting operators to form the chromosome is repeated for a predefined number of iterations (refer to Appendix E for the configuration of LAHC). Given that we have only two mutations per each search iteration, the entire chromosome is not significantly affected. Based on trial runs, we determined a budget of 20 search iterations to provide decent improvement alongside reducing the search cost. Each iteration consists of training the network using the selection activation functions and measuring the training loss $\mathcal{L}_{train}$ as a fitness metric that needs to be minimized.

A downside of this process is the need to retrain the network for each search iteration, which can be intensive in time and compute resources. We circumvent this issue by leveraging a lower fidelity estimation of the final performance. Given that the network performance does not vary after a certain number of epochs, we leverage the work by Loni et al. (2020) and only train the network up to a certain point after which the performance should remain stable.

**Stage 2: Scaling Factor and HPO** Given a learned sequence of optimal operators $\psi$, the next step is to find a sequence $\psi' = \langle (\alpha_i, \beta_i) \mid \alpha_i, \beta_i \in \mathbb{R} \rangle_{i \in \{1,...,L\}}$ representing the scaling factors for each layer. We perform this process jointly with the fine-tuning stage (Equation (3)) and HPO to discover the fine-tuning parameters $\boldsymbol{\theta}^\star_{fine}$ and hyperparameters $\boldsymbol{\lambda}^\star$ as shown in Equation (6).

$$\lambda^{\star} \in \underset{\lambda \in \Lambda}{\arg\min}\, c(\lambda; \mathcal{D}_{val}) \quad \text{s.t.} \quad \psi'^{\star}, \theta_{fine}^{\star}(\lambda) \in \underset{\theta \in \Theta, \psi' \in \mathcal{R}^{(2,L)}}{\arg\min} \left[ \mathcal{L}_{fine}((\theta \mid \theta_{pre}^{\star} \odot m^{\star}), \psi'; \psi, \mathcal{D}_{train}) \right] \quad (6)$$

Due to the continuous nature of this stage, we use the Stochastic Gradient Descent (SGD) for solving Equation (6), and use the validation accuracy as a fitness metric for the hyperparameter configuration.

Treating the scaling factors as learnable parameters allows us to learn them during the fine-tuning state. Thus, the inner optimization in this step has nearly no overhead costs. The only additional cost is that of HPO, which we demonstrate in our experiments to be important and worth it since the hyperparameters from training the original model might not be optimal for fine-tuning.

## 5 Experiments

We categorize the experiments based on the research questions this work aims to answer. Section 5.1 introduces the experimental setup. Section 5.2 motives the problem of tuning activation functions for SNNs. Section 5.3 introduces the need for HPO with activation tuning for SNNs. In Section 5.4, we compare SAFS against different baselines. Appendix D provides an accuracy improvement vs. compression ratio trade-off to compare SAFS with state-of-the-art network compression methods. In Section 5.5 we compare the performance of SAFS for various pruning ratios. In Section 5.6 we provide insights on the activation functions learned by SAFS. Finally, we ablate SAFS in Section 5.7 to determine the impact of different design choices.

### 5.1 Experimental Setup

**Datasets**. To evaluate SAFS, we use MNIST (LeCun et al., 1998), CIFAR-10 (Krizhevsky et al., 2014) and ImageNet-16 (Chrabaszcz et al., 2017) public classification datasets. Note that ImageNet-16 includes all images of the original ImageNet dataset, resized to 16×16 pixels. All HPO experiments were conducted using SMAC3 (Lindauer et al., 2022). Appendix E presents the rest of the experimental setup.

### 5.2 The Impact of Tweaking Activation Functions on the Accuracy of SNNs

To validate the assumption that activation functions indeed impact the accuracy, we investigated whether activation functions currently used for dense networks (Evci et al., 2022) are still reliable in the sparse context. Figure 4a shows the impact of seven different activation functions on the accuracy of sparse architectures with various pruning ratios. To measure the performance during the search stage, we use a three-fold validation approach. However, we report the test accuracy of SAFS to compare our results with other baselines.

Our conclusions from this experiment can be summarised as follows: (i) ReLU does not perform the best in all scenarios. We see that SRS, Swish, Tanh, Symlog, FLAU, and PReLU outperform ReLU on higher sparsity levels. Thus, the decision to use ReLU unanimously can limit the potential gain in accuracy. (ii) As we increase the pruning ratio to 99% (extremely sparse networks), despite the general drop in accuracy, the difference in the sparse and dense networks' accuracies vary greatly depending on the activation function. Thus, the choice of activation function for highly sparse networks becomes an important parameter. We need to mention that despite the success of SAFS in providing higher accuracy, it needs 47 GPU hours in total for learning activation functions and optimal HPs. On the other hand, refining a sparse neural network takes $\approx$ 3.9 GPU hours.

### 5.3 The Difficulty of Training Sparse Neural Networks

Currently, most algorithms for training sparse DNNs use configurations customized for their dense counterparts, e.g., starting from a fixed learning scheduler. To validate the need for optimizing the

training hyperparameters of the sparse networks, we used the dense configurations as a baseline against hyperparameters learned by an HPO method. Figure 4b shows the curves of fine-tuning sparsified VGG-16 with 99% pruning ratio trained on CIFAR-10. The training has been performed with the hyperparameters of the dense network (Blue), and training hyperparameters optimized using SMAC3 (Orange).

We optimized the learning rate, learning rate scheduler, and optimizer hyperparameters with the range specified in Appendix E (Table 4). The type and range of hyperparameters are selected based on recommended ranges from deep learning literature (Simonyan and Zisserman, 2014; Subramanian et al., 2022), SMAC3 documentation (Lindauer et al., 2022), and from the various open-source libraries[4] used to implement VGG-16. To prevent overfitting on the test data, we optimized the hyperparameters on validation data and tested the final performance on the test data. The poor performance (7.17% accuracy reduction) of the SNN learning strategy using dense parameters motivates the need for a separate sparsity-aware HPO regime.

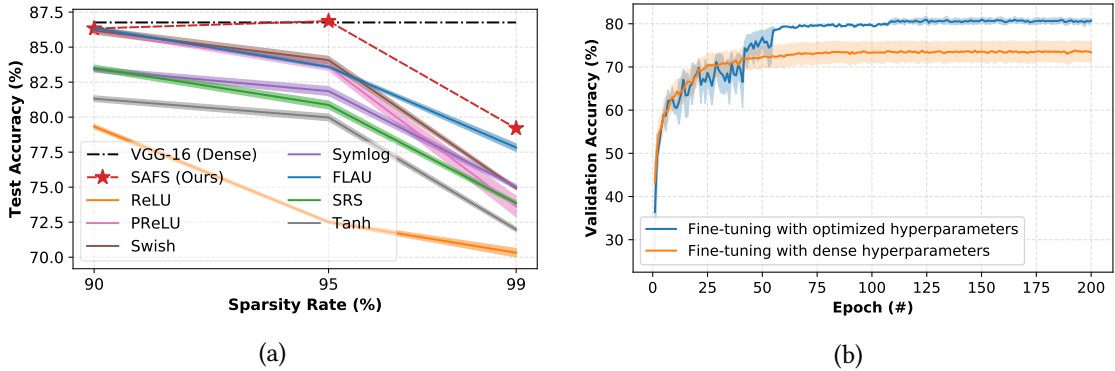

Figure 4: (a) CIFAR-10 test accuracy on sparse VGG-16 with various activation functions customized for dense networks with a 3-fold cross-validation procedure. The bold line represents the mean across the folds, while the shaded area represents the Confidence Intervals across the folds. (b) Fine-tuning sparse VGG-16 on CIFAR-10 with different training hyperparameters with three different random seeds. The pruning ratio is 99%. As shown, fine-tuning with dense hyperparameters results in inefficient training of SNNs.

### 5.4 Comparison with Magnitude Pruning Baselines

Table 1 shows the results of optimizing sparse VGG-16 activation functions trained on CIFAR-10 using SAFS with 99% pruning ratio. An average of three runs has been reported. Results show that SAFS provides 8.88% absolute accuracy improvement for VGG-16 and 6.33% for ResNet-18 trained on CIFAR-10 when compared against a vanilla magnitude pruning baseline. SAFS additionally yields 1.8% absolute Top-1 accuracy improvement for ResNet-18 and 1.54% for EfficientNet-B0 trained on ImageNet-16 when compared against a vanilla magnitude pruning baseline. SReLU (Jin et al., 2016) is a piece-wise linear activation function that is formulated by four learnable parameters. Mocanu et al. (2018); Curci et al. (2021); Tessera et al. (2021) have shown SReLU performs excellently for sparse neural networks due to improving the network's gradient flow. Results show that SAFS provides 15.99% and 19.17% higher accuracy compared to training VGG-16 and ResNet-18 with SReLU activation function on CIFAR-10. Plus, SAFS provides 0.88% and 1.28% better accuracy compared to training ResNet-18 and EfficientNet-B0 with SReLU activation function on the ImageNet-16 dataset. Lastly, Appendix B shows that SAFS significantly improves the gradient flow of sparse neural networks, which is associated with optimized activation functions and efficient training protocol.

---

[4]https://www.kaggle.com/datasets/keras/vgg16/code

Table 1: Refining sparse neural network activation functions with different methods.

| Magnitude Pruning (Han et al., 2015) | CIFAR-10 (Top-1) | | ImageNet-16‡ (Top-1 / Top-5) | |
|---|---|---|---|---|
| | VGG-16 | ResNet-18 | ResNet-18 | EfficientNet-B0 |
| Original Model (Dense) | 86.76% | 89.86% | 25.42% / 47.26% | 18.41% / 37.45% |
| Vanilla Pruning (Baseline) | 70.32% | 77.55% | 11.32% / 25.59% | 10.96% / 25.62% |
| SReLU | 63.21% | 64.71% | 12.24% / 26.89% | 11.22% / 25.98% |
| SAFS (Ours) | 79.2% (+8.88%) | 83.88% (+6.33%) | 13.12% (+1.8%) / 28.94% | 12.5% (+1.54%) / 27.15% |

‡ The Top-1 accuracy of WideResNet-20-1 on ImageNet-16 is 14.82% (Chrabaszcz et al., 2017).

## 5.5 Evaluation of SAFS with Various Pruning Ratios

Figure 4a compares the performance of VGG-16 fine-tuned by SAFS and the default training protocol on CIFAR-10 over three different pruning ratios including 90%, 95%, and 99%. Results show that SAFS is extremely effective by achieving 1.65%, 7.45%, and 8.88% higher accuracies compared to VGG-16 with ReLU activation functions fine-tuned with the default training protocol at 90%, 95%, and 99% pruning ratios. Plus, SAFS is better than activation functions designed for dense networks, especially for networks with a 99% pruning ratio.

## 5.6 Insights on Searching for Activation Functions

Figure 5 presents the dominance pattern of each unary operator in the first learning stage ($\alpha = \beta = 1$) for the CIFAR-10 dataset. The results are the average of three runs with different random seeds. The unit of the color bar is the number of seeing a specific activation function across all search iterations for the first learning stage. According to the results, it is evident that (i) Symexp and ELU are unfavorable activation functions, (ii) Symlog and Acon are dominant activation functions while being used in the early layers, and (iii) Overall Swish and HardSwish are good, but they mostly appear in the middle layers.

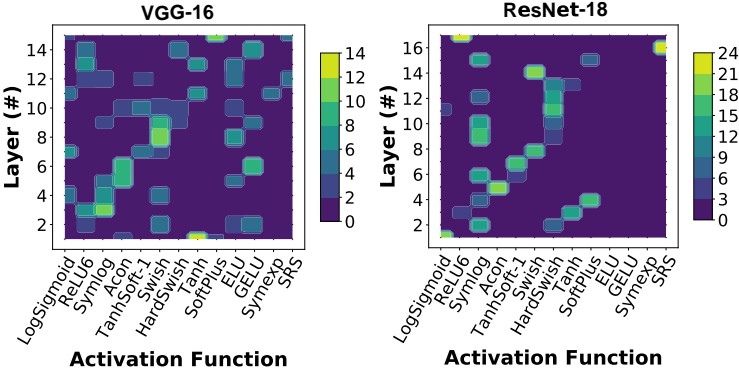

Figure 5: Frequency of Occurring unary operator in the first learning stage ($\alpha = \beta = 1$) for VGG-16 and ResNet-18 trained on CIFAR-10 with 99% pruning ratio.

## 5.7 Ablation Study

We study the effect of each individual optimization stage of SAFS on the performance of sparse LeNet-5, VGG-16, ResNet-18, and EfficientNet-B0 in Table 2. Results show that each individual contribution provides higher accuracy for both VGG-16 and ResNet-18. However, the maximum performance is attained by SAFS (+15.53%, +8.88%, +6.33%, and +1.54% for LeNet-5, VGG-16, ResNet-18, and EfficientNet-B0), where we first learn the most accurate unary operator for each layer and then fine-tune scaling factors with optimized hyperparameters.

Table 2: Ablation Study on optimizing activation functions of SNNs with 99% pruning ratio.

| CNN Model* | Dense Model | Magnitude Pruning | Learning Activation Functions‡ | | |
|---|---|---|---|---|---|
| | | | (Stage 1)† | (Stage 2)‡ | SAFS (Stage 1 + Stage 2) |
| LeNet-5 | 98.49% | 46.69% | 61.63% | 60.2% | 62.22% (+15.53%) |
| VGG-16 | 86.76% | 70.32% | 78.11% | 80.97% | 79.2% (+8.88%) |
| ResNet-18 | 89.86% | 77.55% | 79.34% | 82.74% | 83.88% (+6.33%) |
| EfficientNet-B0 | 18.41% | 10.96% | 11.84% | 11.7% | 12.5% (+1.54%) |

\* Lenet-5, VGG-16, ResNet-18, and EfficientNet-B0 are trained on MNIST, CIFAR-10, CIFAR-10, and ImageNet-16, respectively.
‡ ReLU is the default activation function for Lenet-5, VGG-16, and ResNet-18. Swish is the default activation function for EfficientNet-B0.
† Learning activation functions by only using the first stage of SAFS ($\alpha = \beta = 1$ and without using HPO).
‡ Learning $\alpha$ and $\beta$ for the ReLU operator with optimized hyperparameters.

## 6 Conclusion

In this paper, we studied the impact of activation functions on training sparse neural networks and use this to learn new activation functions. To this end, we demonstrated that the accuracy drop incurred by training SNNs uniformly with ReLU for all units can be partially mitigated by a layer-wise search for activation functions. We proposed a novel two-stage optimization pipeline that combines discrete and stochastic optimization to select a sequence of activation functions for each layer of an SNN, along with discovering the optimal hyperparameters for fine-tuning. Our method SAFS provides significant improvement by achieving up to 8.88% and 6.33% higher accuracy for VGG-16 and ResNet-18 on CIFAR-10 over the default training protocols, especially at high pruning ratios. Crucially, since SAFS is independent of the pruning algorithm, it can optimize any sparse network.

## 7 Limitations and Broader Impact

**Broader Impact**. The authors have determined that this work will have no negative impacts on society or the environment, since this work does not address any concrete application.

**Future Work and Limitations**. Sparse Neural Networks (SNNs) enable the deployment of large models on resource-limited devices by saving computational costs and memory consumption. In addition, this becomes important in view of decreasing the carbon footprint and resource usage of DNNs at inference time. We believe this opens up new avenues of research into methods that can improve the accuracy of SNNs. We hope that our work motivates engineers to use SNNs more than before in real-world products as SAFS provides SNNs with similar performance to dense counterparts. Some immediate directions for extending our work are (i) leveraging the idea of accuracy predictors (Li et al., 2023) in order to expedite the search procedure. (ii) SNNs have recently shown promise in application to techniques for sequential decision-making problems such as Reinforcement Learning (Vischer et al., 2022; Graesser et al., 2022). We believe incorporating SAFS into such scenarios can help with the deployability of such pipelines.

SAFS has been evaluated on diverse datasets, including MNIST, CIFAR-10, and ImageNet-16, and various network architectures such as LeNet-5, VGG-16, ResNet-18, and EfficientNet-B0. While the current results demonstrate the general applicability of our method and signs of scalability, we believe further experiments on larger datasets and more scalable networks would be an interesting avenue for future work.

## Acknowledgements

Aditya Mohan and Marius Lindauer were supported by the German Federal Ministry of the Environment, Nature Conservation, Nuclear Safety and Consumer Protection (GreenAutoML4FAS project no. 67KI32007A). Mohammad Loni was supported by the HiPEAC project, a European Union's Horizon 2020 research and innovation program under grant agreement number 871174.

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

## A Evaluation of Various Search Algorithms

Figure 6 shows the trend of search performance for finding the best unary operators (Equation (5)) over popular search algorithms, including Late-Acceptance-Hill-Climbing (LAHC), Simulated Annealing (SA), Random Search (RS), and Bayesian Optimization (BO). VGG-16 is trained on CIFAR-10 with a 99% pruning ratio. The bold line represents the mean across three random seeds, while the shaded area represents the confidence intervals. Overall, the observation is that SAFS's search algorithm, LAHC, finds better activation functions than other counterparts with an equal search budget.

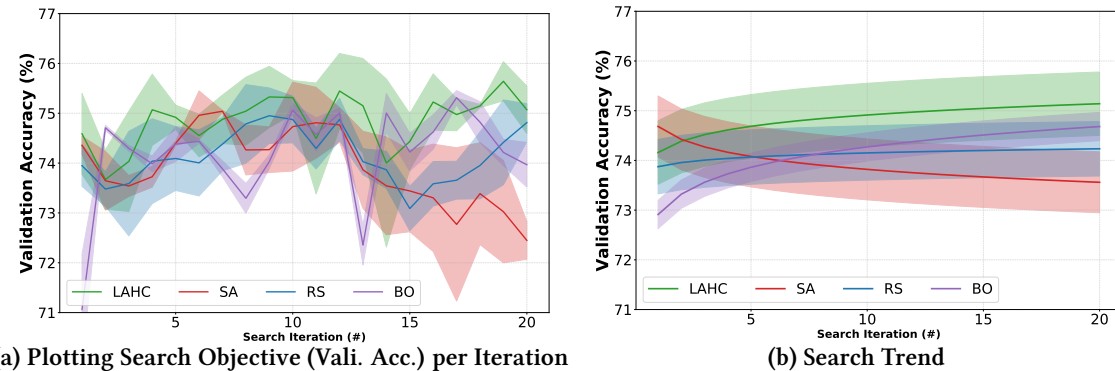

(a) Plotting Search Objective (Vali. Acc.) per Iteration    (b) Search Trend

Figure 6: Comparison of different search algorithms (LAHC, SA, RS, BO) for finding the best unary operators for sparse VGG-16 with 99% pruning ratio trained on CIFAR-10. The bold line represents the mean across three random seeds, while the shaded area represents the confidence intervals. (a) Showing raw data. (b) Using a smoothing average function (logarithmic) for representing the trend of data.

## B Comparing Gradient Flow of SAFS with the Vanilla Pruning

Figure 7 compares the gradient flow of the sparse VGG-16 trained on CIFAR-10 using SAFS (Blue and the vanilla pruning (Orange). As a reminder, gradient flow is the first-order approximation of the decrease in loss after each gradient step, thus the higher the value is the better. Results show that SAFS significantly improves this metric, which is associated with optimized activation functions and efficient training of sparse neural networks.

## C Reporting the Computing Cost of SAFS

Table 3 compares the computing cost (GPU hours) of refining a sparse neural network with SAFS and default vanilla pruning. Although SAFS is slower than the vanilla pruning method, we need to pay this cost only once. Our results show that the significant improvements achieved by SAFS are worth paying this cost. It is important to note that we have not used any multi-fidelity techniques to speed up the first search stage, which is one reason for our slow speed. The use of search acceleration techniques will be explored in the future.

## D Comparison of Accuracy-Compression Ratio Trade-off with State-of-the-Art

We study the effectiveness of SAFS in comparison with various state-of-the-art sparsification and quantization methods in the context of a trade-off between compression ratio (*x-axis*) and performance improvement (*y-axis*) compared to each method's baseline (Figure 8). We examine VGG-16 and ResNet-18 networks trained on CIFAR-10. Our results reveal that SAFS provides 6.24% higher accuracy and 2.18× more compression ratio for VGG-16 over the best counterparts. SAFS achieves 2.42% higher accuracy than the best counterparts with similar compression ratios for ResNet-18.

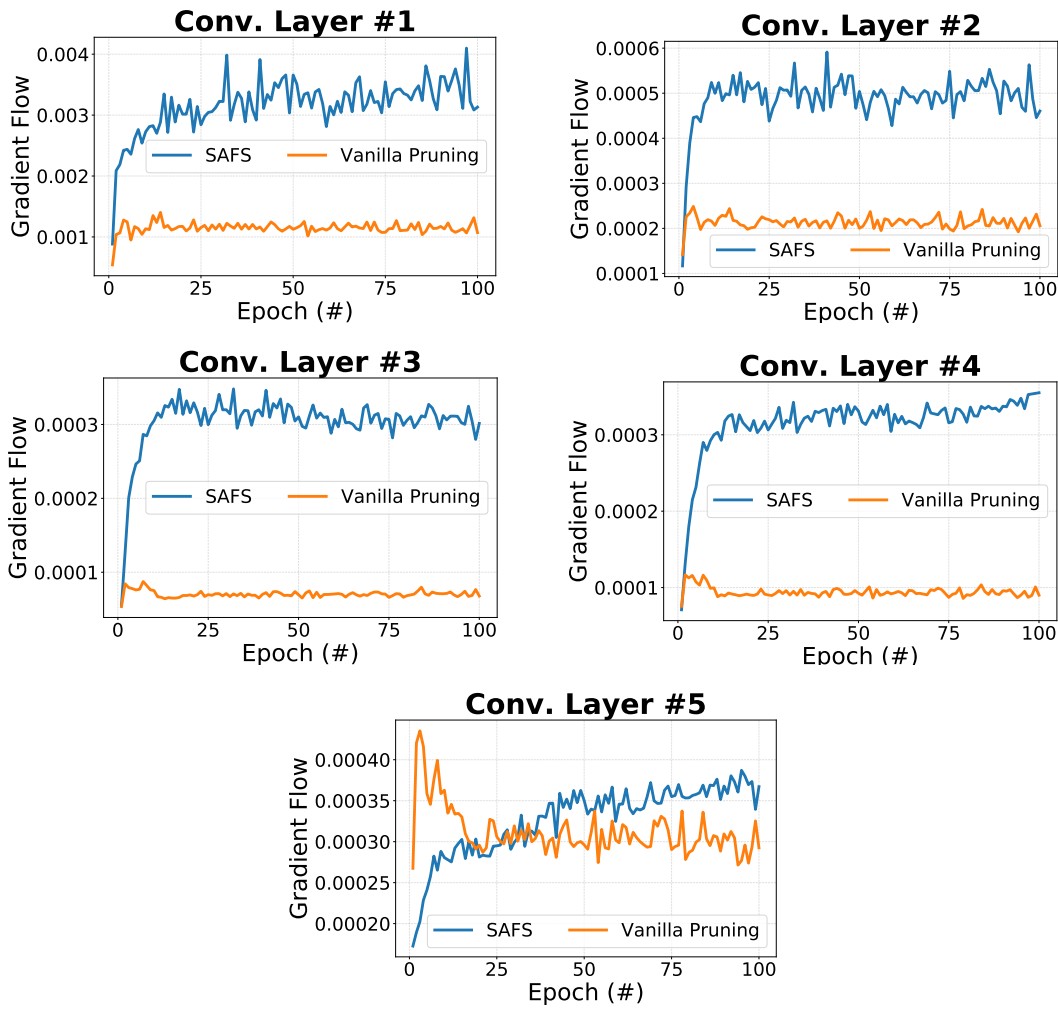

Figure 7: Gradient flow for sparse VGG-16 with 99% pruning ratio with five convolutional layers trained on CIFAR-10. Note that the higher values are the better.

Table 3: Reporting the required computing cost for learning sparse neural network activation functions.

| Network | Dataset | GPU Hours (without considering dense training and sparsification) | |
| | | SAFS (with three-fold cross-validation) | Vanilla Pruning (with one-fold cross-validation) |
|---|---|---|---|
| LeNet-5 | MNIST | 6.4 | 0.16 |
| VGG-16 | CIFAR-10 | 47 | 3.8 |
| ResNet-18 | CIFAR-10 | 63 | 5.6 |
| EfficientNet-B0 | ImageNet-16 | 400 | 7.7 |

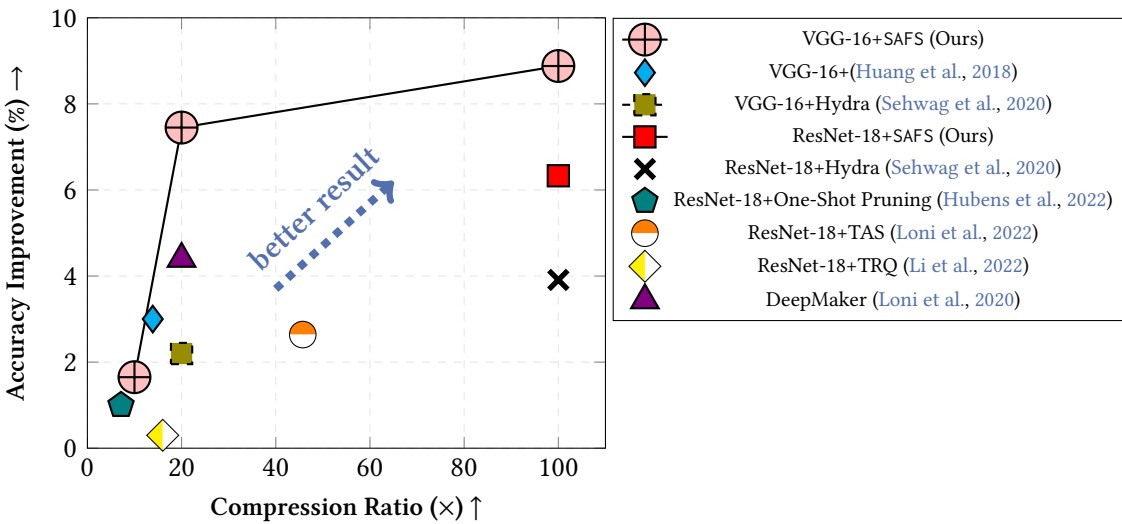

Figure 8: Showing the accuracy improvement (%) vs. the number of network parameters (#Params) of various compact networks trained on CIFAR-10.

## E  Details on Searching Networks

Table 4 shows the configuration details of Stage 1 and Stage 2 learning procedures.

Table 4: Table showing the general hyperparameter configuration for SAFS learning procedures.

| Stage 1: Learning Unary Operators | |
|---|---|
| Unary Operators[‡] | ReLU6 (Howard et al., 2017), Acon (Ma et al., 2021), TanhSoft-1 (Biswas et al., 2021) |
| | SRS (Zhou et al., 2020), Symlog (Hafner et al., 2023), Symexp (Hafner et al., 2023) |
| | Swish, Tanh, HardSwish, ELU, GELU, Softplus, LogisticSigmoid |
| History Length | 3 |
| Number of Iterations | 20 |
| Epochs for Evaluation | 80 |
| Stage 2: Scaling factors and HPO | |
| HPO Library | SMAC3[*] |
| Learning Rate | $1e^{-4} < lr < 1e^{-1}$ |
| Learning Rate Scheduler | *constant, step, linear, cosine annealing (Loshchilov and Hutter, 2017)* |
| | $\{0.001 \times (0.5^{epoch\%20})\}$, *ReduceLROnPlateau[†], CosineAnnealingWarmRestarts[‡]* |
| Optimizer | SGD, Adam, Fromage, TAdam (Ilboudo et al., 2020) |

[‡] (Dubey et al., 2022) explains in detail popular activation functions considered in this study.

[*] https://github.com/automl/SMAC3

[†] https://pytorch.org/docs/stable/generated/torch.optim.lr_scheduler.ReduceLROnPlateau.html

[‡] https://pytorch.org/docs/stable/generated/torch.optim.lr_scheduler.CosineAnnealingWarmRestarts.html

Table 5 provides the configuration details for training the dense LeNet-5 model (baseline) with ReLU activation functions trained on MNIST.

Table 6 provides the configuration details for training dense models (baseline) with ReLU activation functions trained on CIFAR-10.

Table 7 provides the configuration details for training dense models (baseline) with ReLU activation functions trained on ImageNet-16.

Table 8 presents specifications of hardware devices utilized for evaluating the performance of SAFS.

Table 5: Dense CNNs with training hyperparameters for MNIST dataset used in experiments.

| Network[‡] | LeNet-5 |
|---|---|
| Epoch (#) | 100 |
| Learning Rate ($lr$) | 0.1 |
| Learning Rate Scheduler | None |
| Optimizer | SGD |
| Train Time (GPU Hours) for One Model (One-fold) | 0.16 |

[‡] Original implementation of dense model: https://github.com/ChawDoe/LeNet5-MNIST-PyTorch/blob/master/train.py

Table 6: Dense CNNs with training hyperparameters for CIFAR-10 dataset used in experiments.

| Network | VGG-16 | ResNet-18 |
|---|---|---|
| Epoch (#) | 200 | 200 |
| Learning Rate ($lr$) | 0.001 | 0.01 |
| Learning Rate Scheduler | $0.001 \times (0.5^{epoch\%20})$ | $ReduceLROnPlateau^{‡}$: {factor: 0.05, patience: 2, min_lr: 0, threshold: 0.0001, eps:$1e^{-8}$ } |
| Weight Decay | $5e^{-4}$ | $5e^{-4}$ |
| Momentum | 0.9 | 0.9 |
| Optimizer | SGD | SGD |
| Train Time (GPU Hours) for One Model (One-fold) | 1.25 | 4.0 |

[‡] https://pytorch.org/docs/stable/generated/torch.optim.lr_scheduler.ReduceLROnPlateau.html

Table 7: Dense CNNs with training hyperparameters for ImageNet-16 dataset used in experiments.

| Network | EfficientNet-B0 | ResNet-18 |
|---|---|---|
| Epoch (#) | 50 | 50 |
| Learning Rate ($lr$) | 0.01 | 0.1 |
| Learning Rate Scheduler | $CosineAnnealingWarmRestarts^{‡}$: {#Iterations for first restart: 12, Minimum learning rate:$5e^{-5}$ } | $CosineAnnealingWarmRestarts^{‡}$: {#Iterations for first restart: 12, Minimum learning rate:$5e^{-5}$ } |
| Weight Decay | $5e^{-4}$ | $5e^{-4}$ |
| Momentum | 0.9 | 0.9 |
| Optimizer | SGD | SGD |
| Train Time (GPU Hours) for One Model (One-fold) | 18 | 16 |

[‡] https://pytorch.org/docs/stable/generated/torch.optim.lr_scheduler.CosineAnnealingWarmRestarts.html

Table 8: Hardware Specification for search & train.

| Parameter | Specification |
|---|---|
| GPU | NVIDIA® RTX A4000 (735 MHz) |
| GPU Memory | 16 GB GDDR6 |
| GPU Compiler | cuDNN version 11.1 |
| System Memory | 64 GB |
| Operating System | Ubuntu 18.04 |
| $CO_2$ Emission/Day [†] | 1.45 Kg |

[†] Calculated using the ML $CO_2$ impact framework (Lacoste et al., 2019).

