# OpenReview forum: "Learning Activation Functions for Sparse Neural Networks"
_automl.cc/AutoML/2023/Conference — AutoML 2023 MainTrack_

### Official Review · Reviewer_uT44 · 2023-04-12

**Potential Impact On The Field Of Automl Rating:** 3
**Technical Quality And Correctness Rating:** 3
**Clarity Rating:** 3

**Summary Of Contributions:**

In the space of sparse neural networks (SNNs), this paper identifies the lack of SNN-specific development of activation functions among other design choices and proposes Sparse Activation Function Search (SAFS), a method for optimizing layer-wise SNN activation functions in a mixed-integer search space.

**Actions Required To Increase Overall Recommendation:**

* Adding experiments and results requested (in order of importance):
  * Comparison of computational costs for Table 1 and Figure 4(a)
  * Ablation of LAHC vs. other algorithms and of HPO for Table 2
  * Experiments outside of tiny natural image classification
  * Compare to parallel optimization of the unary operator and its parameters
* Expand and clarify analysis in Section 5.6

In response to the additions discussed below and in the general response, I have raised my score from 7 to 8.

**Clarity:**

The paper is very clear overall. The language is easy to understand. Some plots and details are unclear, as detailed below.

**Overall Review:**

**Positive aspects:**

* The problem is motivated clearly.

* Aside from missing experiments below, the authors cover a good range of experiments and results, from explorations to define the problem in Figure 4, main results in Table 1, insightful analysis in Figure 5, and some ablations in Table 2.

**Negative aspects:**

* There is minimal mention (Table 6) nor any comparison of algorithmic cost of SAFS versus baselines and other approaches. This is important information for readers and practitioners to consider the trade-off of computational cost versus performance gain. This is also crucial for Figure 4(a): were all runs given equal computational resources? If not, you need to rerun these experiments but use the equivalent computational cost of SAFS for each baseline model with a single activation function. Computational cost information should also be included at least for all experiments in Table 2.

* The choice of LAHC over other discrete search algorithms is only supported by Figure 6. Figure 6(a) doesn't seem very conclusive due to noise I don't completely understand Figure 6(b): is this the same data but with some sort of smoothed/windowed average? This choice could be further supported by including it in the ablation studies.

* The idea of Section 5.6 seems interesting but not yet completely developed. Do the patterns you noted extend to other backbones and tasks? With only 3 runs represented in Figure 5 (and only their averages, no measures of variance), it's not clear how much signal there is compared to noise. Also, I don't understand what is being plotted: what are the units of the colorbar?

* In Section 5.2, it's not clear what you are using "three-fold validation" here for, since you are comparing networks with different activation functions (with no inner optimizations) and Figure 4(a) shows test accuracy, not validation accuracy.

* The breadth of backbones and tasks is relatively narrow. How does this method fair in other modalities, network types, etc.? Can the same unary operator search space be used, or does it need to be adapted to the backbone and input modality?

**Further Experiments:** (could be included in follow-up work)

* Your ablation studies are a good start, but since Figure 4(b) shows significant gain from just HPO, I would like to also see an ablation that only removes HPO (as well as an ablation that only includes HPO).

* Your search space is well divided into selecting a unary operator and it's two continuous parameters. Your algorithm approaches these decisions in series, but I think tackling them in parallel could be more effective. Instead of having two stages, instead initialize $\alpha$ and $\beta$ for each activation function and each layer. Then, the matrices containing these parameters could be optimized by SGD simultaneously to the discrete optimization of $\Psi$.


**Minor comments:**

* Line 69: "A number of studies have proposed the necessity of pruning..." - the word "necessity" doesn't feel right here. Perhaps "...have proposed to prune..." instead?

* Figures 1 and 2 appear out of order.

* For Figure 5, right-align the bottom labels and rotate them by the right side, rather than the center, using something like this:
```plt.setp(axs[i].get_xticklabels(), rotation=45, ha="right", rotation_mode="anchor")```

**Potential Impact On The Field Of Automl:**

This paper has medium potential impact on the field of AutoML. It applies existing ideas such as LAHC and SMAC3 towards the novel application of activation function design for SNNs, which can be used in an AutoML pipeline. This paper may be cited in further activation function design studies that expand the search space or in papers that study other aspects of SNN design.

**Review Confidence:**

4: You are confident in your assessment, but not absolutely certain. It is unlikely, but not impossible, that you did not understand some parts of the submission or that you are unfamiliar with some pieces of related work.

**Review Rating:**

8: Accept: Technically sound paper with major impact and strong evaluation, with perhaps some minor flaws.

**Review Summary:**

This paper is a clear and impactful study on activation function optimization for SNNs. It could be further supported by additional experiments detailed above. I think it is worth accepting, especially if the recommended actions are taken.

**Technical Quality And Correctness:**

The breadth of results is good overall, although some additional experiments and analyses are recommended below. The introduction and related works are good to my knowledge.

---

> ### Author Response · Authors · 2023-04-28
> **Response to Reviewer uT44**
>
> ## Overall Review
>
> > There is minimal mention (Table 6) nor any comparison of algorithmic cost of SAFS versus baselines and other approaches.
>
> In Appendix B, we have added a discussion of the cost of using SAFS in terms of GPU hours.  We also reported the GPU cost for optimizing activation functions of VGG-16 (Figure 1) at the end of Section 5.2. We admit that SAFS adds a substantial overhead, but we believe that this will be compensated by the downstream application of these models.
>
> > The choice of LAHC over other discrete search algorithms is ...
>
> Figure 6(a): Although we have small noise, LAHC showed promising results across all search iterations compared to other search methods.  Figure 6(b): Sorry for being unclear. It shows the same data, but we used a smoothing average (logarithmic function) to show the trend in the data more clearly.  We have added a statement regarding thisin the paper
>
> > The idea of Section 5.6 seems interesting but not yet completely developed. Do the patterns you noted ...
>
> The unit of the color bar is the number of seeing a specific activation function across all search iterations (stage 1). Overall, activation functions with higher values are potentially more promising. Note that we have added this explanation to the paper. The idea of Figure 5 is to give some insights to the community and researchers who want to extend our work. Our results show that, for example, SRS, GELU, and Symexp activation functions have less chance to be selected.
>
> > In Section 5.2, it's not clear what you are using "three-fold validation" here for, since you are comparing networks with different activation functions (with no inner optimizations) and Figure 4(a) shows test accuracy, not validation accuracy.
>
> We wanted to use a statistically robust way with multiple train/validation splits via 3-fold cross-validation instead of evaluating the performance on a single train/validation set. In addition, during the search time, we use the validation set to evaluate candidates. However, in Figure 4(a), we have reported the final test accuracy results to be comparable with other baselines. We have revised the manuscript to make it clear.
>
> > The breadth of backbones and tasks is relatively narrow. How does this method fair in other modalities, network types, etc.? Can the same unary operator search space be used, or does it need to be adapted to the backbone and input modality?
>
> Our proposed method is agnostic to the network architecture. For example, we could in principle apply SAFS to optimize the activation parts of vision transformers, such as encoder networks. There is no further adaptation needed to the backbone or other modalities. If further activation function classes should be needed, they can easily be added to the search space of SAFS.
>
> > Your ablation studies are a good start, but since Figure 4(b) shows significant gain from just HPO, I would like to also see an ablation that only removes HPO (as well as an ablation that only includes HPO).
>
> We have updated the ablation results (Table 2) by adding the results of EfficientNet-B0 trained on ImageNet-16 and LeNet-5 trained on MNIST. The Stage 1 column in the table shows the results of the activation function search by removing HPO (dense HPs). The Stage 2 column includes the HPO results without optimizing activation functions (using the default activation function for the network). Finally, we report the results of our proposed method, SAFS=Stage 1 + Stage 2, in the last column.
>
> > Your search space is well divided into selecting a unary operator and it's two continuous parameters. Your algorithm approaches these decisions in series, but I think tackling them in parallel could be more effective. Instead of having two stages, instead initialize 𝜶 and 𝜷 for each activation function and each layer. Then, the matrices containing these parameters could be optimized by SGD simultaneously to the discrete optimization of Ψ.
>
> Thank you very much for the great suggestion. We will consider working on this idea in future works.
>
> We addressed all the minor comments in the new revision of our paper.
>
> ## Actions Required To Increase Overall Recommendation:
>
> > 1. Adding experiments and results requested (in order of importance):
> > * Comparison of computational costs for Table 1 and Figure 4(a).
>
> Addressed by Appendix B.
>
> > * Ablation of LAHC vs. other algorithms and of HPO for Table 2
>
> We have updated Table 2.
>
> > * Experiments outside of tiny natural image classification
>
> Remain to future work.
>
> > * Compare to parallel optimization of the unary operator and its parameters
>
> Remain to future work.
>
> > 2. Expand and clarify analysis in Section 5.6
> We have added a small explanation to clarify Section 5.6.

---

> > ### Comment · Reviewer_uT44 · 2023-05-01
> > **Response to authors**
> >
> > I appreciate the updates. I will raise my score from a 7 to an 8.

---

### Official Review · Reviewer_ziJv · 2023-04-12

**Potential Impact On The Field Of Automl Rating:** 3
**Technical Quality And Correctness Rating:** 3
**Clarity Rating:** 3

**Summary Of Contributions:**

This paper proposes SAFS, an automated search for activation functions for sparse neural networks in conjunction with hyperparameter optimization. This method is a two-step procedure, which uses the already pruned neural network as starting and initialization point; first, the activation function for each layer in the network is searched using Late Acceptance Hill Climbing. Second, the hyperparameter for this identified network is optimized.  This paper evaluates SAFS on three different common CNNs (ResNet-18, VGG16, and EfficientNet-B0) for CIFAR-10 and ImageNet16 image classification and shows improved results over pruning baselines, which use the default HP of the dense network.

**Actions Required To Increase Overall Recommendation:**

Additional experiments regarding the scalability of the proposed method or at least a discussion:

-	Larger networks like ResNet-50.
-	Larger Datasets with larger resolutions, Imagenette or ImageNet100

Different activation functions as ablation for the used dense networks.

**Post-Rebuttal:**
I updated my score from 6 to 7.

**Clarity:**

This paper is clear and motivates the research properly.

There are some open questions and remarks, which would further improve the clarity:
Line 12: “learning novel activation functions for sparse networks”: This sentence can be misleading, since this paper does not propose a truly novel activation function, but rather searches for activation functions at each layer.

Figure 2 is not referenced.

LAHC is not clearly described. Therefore, more background on that is helpful, since lines 190-195 are not comprehensible for readers not being familiar with LAHC.

Lines 217-220: scaling factors of activation functions are trainable parameters, and not searched by means of HPO. Therefore, the sentence in the mentioned lines might be misleading

Section 5.6 is not mentioned in the introductory part of Section 5.

The overall used methods could be mentioned, SMAC3 was only mentioned in line 253. So is SMAC3 used for HPO?

Mentioning the default activation functions for the CNNs would be helpful since EfficientNet-B0 used Swish instead of ReLU.

A comparison for each used CNN with different activation functions for the default dense network would provide a stronger starting point (see Technical Quality And Correctness)

Table 1: How many runs were conducted?

Section 5.5 rather repeats the information from Section 5.2. These could be combined.

Figure 5: better alignment of the x-axis labels for better readability.


**Overall Review:**

Strengths:
This paper proposes an interesting research question with sound experiments.
Good motivation and clear assessment of why this is important (given the studies in Sections 5.1 and 5.2)
The idea to search for activation functions and HPO in conjunction is novel.

Weaknesses:
How does this method scale to larger networks, e.g., ResNet-50, and larger image resolutions (Imagenette, ImageNet 100) with the focus on how the input image resolutions influence this method.


**Potential Impact On The Field Of Automl:**

This work has a medium impact on the field of AutoML as it proposed an approach limited to only sparse neural networks. However, this area is receiving more and more focus, aiming for less resource usage.


**Review Confidence:**

4: You are confident in your assessment, but not absolutely certain. It is unlikely, but not impossible, that you did not understand some parts of the submission or that you are unfamiliar with some pieces of related work.

**Review Rating:**

7: Weak Accept: Technically sound paper with moderate-to-high impact and strong evaluation, with perhaps some minor flaws.

**Review Summary:**

While the idea of combining activation functions and HPO for sparse neural networks is novel and great, I believe some additional assessments and experiments are useful to improve this paper.



**Technical Quality And Correctness:**

The conducted experiments are appropriate and sound and support the motivation of this paper as well as the content in general.

However, in my opinion, there are still some missing experiments and ablations:

Section 5.2: What is the influence of the different activation functions at sparsity level 0%, i.e. dense network? (see Figure 4a)

Table 2: For VGG-16 it seems to be the case that either the different activations at each layer improve the accuracy, or the HP itself, while the combination of both even decreases the accuracy. While for ResNet-18 SAFS improves the accuracy the most. Why is that so?
Also, how does EfficientNet-B0 behave in this ablation?

The overall search times are missing.

---

> ### Author Response · Authors · 2023-04-28
> **Response to Reviewer ziJv**
>
> ## Technical Quality And Correctness:
> > Section 5.2: What is the influence of the different activation functions at ...
>
> The focus of this paper is to improve the accuracy of SNNs by jointly learning network activation functions and HPs. Several prior studies proposed to optimize activation functions [Ref7][Ref8][Ref9] and/or HPs of dense networks (0% sparsity level) [Ref10]. From our perspective, these are orthogonal to our work since we specifically focus on the sparse setting.
> * [Ref7] Apicella, Andrea, et al. "A survey on modern trainable activation functions." Neural Networks 138 (2021): 14-32.
> * [Ref8]  Dubey, Shiv Ram, Satish Kumar Singh, and Bidyut Baran Chaudhuri. "Activation functions in deep learning: A comprehensive survey and benchmark." Neurocomputing (2022).
> * [Ref9] Karlik, Bekir, and A. Vehbi Olgac. "Performance analysis of various activation functions in generalized MLP architectures of neural networks." International Journal of Artificial Intelligence and Expert Systems 1.4 (2011): 111-122.
> * [Ref10] Yu, Tong, and Hong Zhu. "Hyper-parameter optimization: A review of algorithms and applications." arXiv preprint arXiv:2003.05689 (2020).
>
> > Table 2: For VGG-16 it seems to be the case that either the different activations at each layer improve the accuracy, or ...
>
> We have updated the ablation results (Table 2) by adding the results of EfficientNet-B0 trained on ImageNet-16 and LeNet-5 trained on MNIST.  Results show that using only HPO does not always provide the best results. For the MNIST and ImageNet-16 datasets, we can achieve 13.51% and 0.74% accuracy improvements by using HPO. While using efficient activation functions provides us with up to 2.02% and 0.55% higher accuracy than only using HPO.
>
> > The overall search times are missing.
>
> Appendix B reports the cost of using SAFS for optimizing different networks.
>
> ## Clarity:
> > Line 12: “learning novel activation functions for sparse networks”: This sentence can be misleading, since ...
>
> We have changed this to ‘learning a novel way to tune activation functions for sparse networks’ for better clarity.
>
> > Figure 2 is not referenced.
>
> We have referenced Figure 2 in the text.
>
> > LAHC is not clearly described. Therefore, more background on that is helpful, since lines 190-195 are not ...
>
> We have added a smaller explanation of LAHC to make this more self-contained.
>
> > Lines 217-220: scaling factors of activation functions are trainable parameters, and ...
>
> We have updated the sentence to better reflect your point.
>
> > Section 5.6 is not mentioned in the introductory part of Section 5.
>
> We have added a sentence for this.
>
> > The overall used methods could be mentioned, SMAC3 was only mentioned ...
>
> Yes, SMAC3 is used for HPO in all experiments. We have added a small explanation to  Section 5.
>
> > Mentioning the default activation functions for the CNNs would be helpful since ...
>
> Sorry for being unclear. The default activation function for EfficientNet-B0 is Swish. We have mentioned it in Table 2.
>
> > A comparison for each used CNN with different activation functions ...
>
> The default activation functions of each dense network are the best choice that is designed by experts. In addition, regarding dense networks, many related studies [Ref7][Ref8][Ref9] well-studied the impact of tweaking activation functions. And the focus of this paper is on sparsity cases.
>
> > Table 1: How many runs were conducted?
>
> We used three runs. We have added an explanation to Section 5.4.
>
> > Section 5.5 rather repeats the information from Section 5.2.
>
> Even though a small part of the results is redundant, we aim to point to a specific conclusion in each section.
>
> > Figure 5: better alignment of the x-axis labels for better readability.
>
> We have fixed the problem.
>
>
> ## Actions Required To Increase Overall Recommendation:
>
> > Additional experiments regarding the scalability of the proposed method or at least a discussion:
>
> SAFS has been evaluated on diverse datasets, including MNIST, CIFAR-10, and ImageNet-16, and various network architectures such as LeNet-5, VGG-16, ResNet-18, and EfficientNet-B0. While the current results demonstrate the general applicability of our method and signs of scalability, we believe further experiments on larger datasets and more scalable networks would be an interesting avenue for future work.
>
> > Different activation functions as ablation for the used dense networks.
>
> The focus of this paper is to improve the accuracy of sparse neural networks by jointly learning network activation functions and HPs. Several prior studies proposed to optimize activation functions [Ref7][Ref8][Ref9] of dense networks. In view of sufficient studies on this topic, we see this as out of scope for our paper and focus on the sparse case.

---

> > ### Comment · Reviewer_ziJv · 2023-05-08
> > **Response to Authors**
> >
> > I thank the authors for their response and update my score from 6 to 7.

---

### Official Review · Reviewer_JFfa · 2023-04-13

**Potential Impact On The Field Of Automl Rating:** 3
**Technical Quality And Correctness:** The claims appear to be correct.
**Technical Quality And Correctness Rating:** 4
**Clarity:** The paper is well-written and easy to…
**Clarity Rating:** 4

**Summary Of Contributions:**

This paper introduces a technique for improving sparse neural networks by optimizing the hyperparameters and activation function in addition to the sparsity mask, which are often ignored in the sparse neural network literature. This search space over activation functions comprises a set of unitary functions with learned scaling of the inputs and outputs, and a hill-climbing algorithm is used to optimize the activation function, with experiments conducted on CIFAR-10 and a resized version of ImageNet.

**Actions Required To Increase Overall Recommendation:**

I would suggest that the authors include at least one additional dataset in their empirical evaluation.

**Overall Review:**

Strengths:
- This combination of research areas appears to be novel and fills an often-overlooked gap in the literature (tuning activations and hyperparameters of sparse neural networks)
- The current set of experimental results seems promising
- The ablation study of the different stages of the proposed search method is a useful experiment to include.

Weaknesses:
- As an empirical work, it would be ideal to experiment with more than two datasets. Currently, only CIFAR-10 and resized ImageNet are evaluated. On the other hand, this is potentially understandable as these experiments are likely rather expensive.


**Potential Impact On The Field Of Automl:**

The sparse neural network literature is rich and in general has a strong potential for impact by reducing computation via sparsity. General methods for improving sparse neural networks similarly share this promise of impact. While this field is related to AutoML, the impact seems specific to sparse neural networks rather than AutoML as a whole, which is still reasonable.

**Review Confidence:**

3: You are fairly confident in your assessment. It is possible that you did not understand some parts of the submission or that you are unfamiliar with some pieces of related work.

**Review Rating:**

6: Borderline Leaning Accept: Technically sound paper where reasons to accept outweigh reasons to reject. Please use sparingly.

**Review Summary:**

The idea is novel and is well explored on the two datasets that are included, CIFAR-10 and resized ImageNet, however, the evaluation could be improved by including more datasets (particularly since this is an empirical paper).

---

> ### Author Response · Authors · 2023-04-28
> **Response to Reviewer JFfa**
>
> > As an empirical work, it would be ideal to experiment with more than two datasets. Currently, only CIFAR-10 and resized ImageNet are evaluated. On the other hand, this is potentially understandable as these experiments are likely rather expensive. I would suggest that the authors include at least one additional dataset in their empirical evaluation.
>
> Thanks for the careful review and very positive feedback. We have extended our experiments by optimizing the sparse LeNet-5 architecture with a 99% pruning ratio trained on the MNIST dataset. Our results show that SAFS improves the absolute classification accuracy of LeNet-5 by 15.53% (Table 2).

---

> > ### Comment · Reviewer_JFfa · 2023-05-09
> > **Thank you for including an additional dataset**
> >
> > Thank you for including a third dataset. I have updated my score from 5 to a 6 and now lean toward acceptance.

---

### Official Review · Reviewer_AwGP · 2023-04-13

**Potential Impact On The Field Of Automl Rating:** 3
**Technical Quality And Correctness Rating:** 3
**Clarity:** Easy to follow.
**Clarity Rating:** 3
**Actions Required To Increase Overall Recommendation:** Please add experimental results to ea…

**Summary Of Contributions:**

This paper proposed to search for activations and tune hyperparameters for sparse neural networks. Searching for layerwise activation functions is a new perspective to improve the performance of sparse networks. It showed the effect of different activation functions.

**Overall Review:**

This paper provides a new perspective to boost the performance of sparse networks. I am not sure if searching for activations really works. I am not sure if we can simply tune parameters to get the similar performance gain as the proposed SAFS.

**Potential Impact On The Field Of Automl:**

I feel it may generate some discussion in the AutoML community. Searching for activation functions to improve performance of sparse networks is a new perspective.

**Reproducibility (Optional):**

Code is provided. Not sure.

**Review Confidence:**

4: You are confident in your assessment, but not absolutely certain. It is unlikely, but not impossible, that you did not understand some parts of the submission or that you are unfamiliar with some pieces of related work.

**Review Rating:**

6: Borderline Leaning Accept: Technically sound paper where reasons to accept outweigh reasons to reject. Please use sparingly.

**Review Summary:**

My initial decision is borderline accept. I raised some concerns, especially if we can simply use HPO to get the similar performance gain as the proposed method. I am not sure if we really need to search for activations for sparse networks. I hold room to decrease scores if my major concerns are not addressed after rebuttal.

**Technical Quality And Correctness:**

[1] Why do you need to formulate the activation function as $y=\alpha f(\beta x)$. The scaling parameters $\alpha$ and $\beta$ should be learned by convolutional layers.

[2] You did not compare your algorithm with a strong baseline. And you search for layerwise activation functions from existing activation functions. [R1] searches for activations from basic primitives. Can you compare your algorithm with [R1]?

[3] In your ablation study (5.7), stage 2 even outperforms SAFS on VGG-16. Does it mean HPO is more useful than searching for activation functions? In Figure 4, it showed ReLU is inferior to other activation functions. Why did you only pick up ReLU in your ablation study? Can you add more experimental results on other activations, like FLAU, Tanh. From your ablation study, I do not know if searching for activation functions is really working. Furthermore, can you provide an ablation study of tuning hyperparameters of naive pruning methods?


[R1] Evolving normalization-activation layers, NeurIPS 2020.

---

> ### Author Response · Authors · 2023-04-28
> **Response to Reviewer AwGP**
>
> We thank you for your constructive feedback and your overall positive evaluation.
>
> > Why do you need to formulate the activation function as y=𝜶×f(𝜷×X). The scaling parameters 𝜶 and 𝜷 should be learned by convolutional layers.
>
> As discussed in Section 4.1, this representation allows efficient implementation. Additionally, this formula provides effective parameterization since tweaking 𝜶 and 𝜷 parameters give the sparse network additional flexibility to be fine-tuned (Figure 1). Our findings demonstrate that the suggested activation function has notably enhanced the test accuracy of sparse neural networks.
>
> > You did not compare your algorithm with a strong baseline. And you search for layerwise activation functions from existing activation functions. [R1] searches for activations from basic primitives. Can you compare your algorithm with [R1]?
>
> Thank you for the feedback. We considered comparing SAFS to two baselines: (1) vanilla pruning (the most popular method for refining pruned networks), and (2) SReLU (suitable for SNNs suggested by several studies [Ref4][Ref5][Ref6]).  [R1] is an evolutionary-based method that jointly designs batch normalization-activation function layers from elementary operators. However, their proposed method is slow with marginal improvements, and the EvoNorm layer is not effective anymore for many networks such as EfficientNet. Thus, [R1] is not a strong baseline from our point of view. Although it is still interesting to see the results of applying [R1] on SNNs, due to the limited time for rebuttal, we cannot compare our work with [R1] at the moment.
>
> * [Ref4] Mocanu, D., Mocanu, E., Stone, P., Nguyen, P., Gibescu, M., and Liotta, A. (2018). Scalable training of artificial neural networks with adaptive sparse connectivity inspired by network science. Nature Communications.
> * [Ref5] Curci, S., Mocanu, D., and Pechenizkiyi, M. (2021). Truly sparse neural networks at scale. CoRR.
> * [Ref6] Tessera, K., Hooker, S., and Rosman, B. (2021). Keep the gradients flowing: Using gradient flow to study sparse network optimization. CoRR.
>
> > In your ablation study (5.7), stage 2 even outperforms SAFS on VGG-16. Does it mean HPO is more useful than searching for activation functions? In Figure 4, it showed ReLU is inferior to other activation functions. Why did you only pick up ReLU in your ablation study? Can you add more experimental results on other activations, like FLAU, Tanh. From your ablation study, I do not know if searching for activation functions is really working. Furthermore, can you provide an ablation study of tuning hyperparameters of naive pruning methods?
>
> According to our new ablation results on MNIST and ImageNet-16 datasets (see Section 5.7), using only HPO does not always provide the best results. For the MNIST and ImageNet-16 datasets, we can achieve 13.51% and 0.74% accuracy improvements by only using HPO. While using efficient activation functions provides us with up to 2.02% and 0.55% higher accuracy than only using HPO. Thus, searching for activation functions adds performance improvement, further motivating the need to do so.
>
> The aim of our ablation study is to individually evaluate the impact of each optimization step. We selected ReLU as the comparison baseline since ReLU is the default activation function for LeNet-5, VGG-16, and ResNet-18 (Swish is the default activation function for EfficientNet-B0). Finally, as can be seen in Figure 4(a), SAFS always provides the best activation functions for SNNs compared to all other state-of-the-art activation functions. Therefore, we expect that adding other dense activation functions to our ablation study will not change the results.
>
> > I raised some concerns, especially if we can simply use HPO to get the similar performance gain as the proposed method. I am not sure if we really need to search for activations for sparse networks.
>
>  We have addressed the concern in response to Question 3.

---

### Review · Reproducibility_Reviewer_ziGT · 2023-04-14

**Completeness Of Code And Dataset Supplement Rating:** 3
**Usability And Ease Of Reproducibility Rating:** 3

**Actions Required To Increase The Reproducibility And Overall Recommendation:**

line 10 of the README says "requirement.txt" but should end with an s "requirements.txt".

line 14 of the readme refers to "train.py", should it be referring to "main.py"?

You should add the specification 'smac==1.4.0' (or some other 1.x version) to your requirements file, as smac without any other reference downloads smac2.x on my machine, and breaks your imports in optimization/smac_opt.py.

When you overview parameters in line 13, perhaps you could point the reader to main.py and settings.py for more detailed information on what, for example, the different model_arch integer parameters mean.

Unfortunately, when running your example script, I get a stacktrace followed by: "RuntimeError: ProcessGroupNCCL is only supported with GPUs, no GPUs found!".
Unforutunately, I do not have access to a GPU, so I was not able to run these models (seems like model_arch=2,3,4).

I encountered the following issues after changing model_arch to 1:
i) The __ini__.py file in models is lacking "from .dense_model_architecture import *".
ii) The error "AttributeError: module 'settings' has no attribute 'DATASET'" eminating from the file models/dense_model_architecture.py on line 67. I was not able to correct this error (first I thought it was just a missing S, as settings.py defines a DATASETS field, but this did not solve the issue.)

I next tried a model_arch of 0, which according to your settings.py file is a 1-Layer-MLP. Unfortunately, I got the error "UnboundLocalError: local variable 'model' referenced before assignment" eminating from the file models/model.py, and inspection of this file revealed that '1-Layer-MLP' is not one of the options that file checks for (and since the conditional statement is missing an "else" statement checking for bad inputs, we observe the UnboundLocalError).


**Completeness Of Code And Dataset Supplement:**

The code appears complete at first glance (though I was unsuccessful in confirming this by running it).

**Overall Reproducibility Review:**

The directory looks nicely organized, but as I mention in further detail below, there are a few small issues to be resolved.

Unfortunately, without access to a GPU, I could not run the larger models, and I encountered errors trying to run the smaller ones. Subsequently, I could not reproduce this work.

**Review Confidence:**

2: You are willing to defend your assessment, but it is quite likely that you did not understand the central parts of the submission or that you are unfamiliar with the code or data.

**Review Rating:**

7: Weak Accept, all critical aspects are reproducibile with minor effort, and the remainder are likely reproducible with minor additional effort.

**Review Summary:**

Due to my computational infrastructure, I was not able to run the more complex models. Due to what seem like minor mistakes in the scripts, I was not able to try the smaller models either. Therefore, I unfortunately was not able to reproduce any aspect of this article, though manual inspection of several of the source files revealed what seems to be well-written and documented code.

I expect that it's just a small issue preventing me from running the simpler examples, and expect to substantially increase my rating when this is possible.

I have listed my review confidence as low because of my inability to reproduce the GPU results, but I am confident in my static assessment of the code/directory structure.

**Summary Of Necessary Code And Dataset Supplement:**

This article advocates to optimize over hyperparameters of the activation function layer-by-layer in a manner specifically tuned for sparse neural networks.

The article relies on existing network architectures, such as VGG-16, ResNet-18, and EfficientNet-B0, as well as existing datasets such as CIFAR-10 and ImageNet-16.

They conduct their actual optimization via a local descent method, only partially training the neural network so as to avoid complete retraining for each hyperparameter configuration.

**Usability And Ease Of Reproducibility:**

Unfortunately, I do not have access to a GPU. Running the example script provided seems to require a GPU.

I tried running it on simpler models by changing the model_arch param to 0/1 which seemed to be simpler models. Unfortunately, other errors, which I detail below in "Actions Required", prevented me from running this script as well. So I was not able to reproduce the results.

However, the README contains easy to follow install instructions (though there are a few problems, see below again), and the directory seems well organized.

---

> ### Author Response · Authors · 2023-04-28
> **Response to Reproducibility Reviewer ziGT**
>
> > line 10 of the README says "requirement.txt" but should end with an s "requirements.txt".
> * Thanks for the feedback. We have fixed the problem.
>
> > line 14 of the readme refers to "train.py", should it be referring to "main.py"?
> * Thanks for the feedback. We have fixed the problem.
>
> > You should add the specification 'smac==1.4.0' (or some other 1.x version) to your requirements file, as smac without any other reference downloads smac2.x on my machine, and breaks your imports in optimization/smac_opt.py.
> * Thanks for the feedback. We have fixed the problem. We have used SMAC==1.4.0.
>
> > When you overview parameters in line 13, perhaps you could point the reader to main.py and settings.py for more detailed information on what, for example, the different model_arch integer parameters mean.
> * Thanks for the feedback. We have added more details on the model architecture selection.
>
> > Unfortunately, when running your example script, I get a stack trace followed by: "RuntimeError: ProcessGroupNCCL is only supported with GPUs, no GPUs found!". Unfortunately, I do not have access to a GPU, so I was not able to run these models (seems like model_arch=2,3,4).
> * To speed up the training of our CNN model, we have developed our code to train models using a GPU.
>
> > I encountered the following issues after changing model_arch to 1:
> i) The __ini__.py file in models is lacking "from .dense_model_architecture import *".
> ii) The error "AttributeError: module 'settings' has no attribute 'DATASET'" emanating from the file models/dense_model_architecture.py on line 67. I was not able to correct this error (first I thought it was just a missing S, as settings.py defines a DATASETS field, but this did not solve the issue.)
> *  We just added the results of SAFS on the LeNet-5 model trained on the MNIST dataset. Now, our code works for the model_arch set to 1.
>
> >I next tried a model_arch of 0, which according to your settings.py file is a 1-Layer-MLP. Unfortunately, I got the error "UnboundLocalError: local variable 'model' referenced before assignment" eminating from the file models/model.py, and inspection of this file revealed that '1-Layer-MLP' is not one of the options that file checks for (and since the conditional statement is missing an "else" statement checking for bad inputs, we observe the UnboundLocalError).
> *  Our code does not work for model_arch:=0 (1-Layer-MLP) and there are no related results in the paper on a 1-layer MLP. We have used 1-Layer-MLP for some debugging purposes. We have removed 1-Layer-MLP and all corresponding settings from our code.
>
> ## Summary
> In summary, we have addressed all the problems with the code that were pointed out by reviewer ziGT. Additionally, we have uploaded the final version of the code to the  AutoML anonymous Git repository.

---

> > ### Comment · Reproducibility_Reviewer_ziGT · 2023-05-05
> > **thanks for the updates**
> >
> > Thank you for making these changes, I have upped my score and decreased my certainty (to reflect my inability to reproduce the GPU results).

---

### Official Review · Reviewer_YVAr · 2023-04-18

**Potential Impact On The Field Of Automl Rating:** 3
**Technical Quality And Correctness Rating:** 2
**Clarity:** NA
**Clarity Rating:** 3
**Actions Required To Increase Overall Recommendation:** see Technical Quality And Correctness…

**Summary Of Contributions:**

This paper proposed a Sparse Activation Function Search (SAFS) method for optimizing the activation functions and training hyperparameters of Sparse Neural Networks (SNNs). The authors show that the accuracy drop in SNNs, especially at high pruning ratios, can be attributed to the use of ReLU as the default activation function and the use of the same hyperparameters as dense counterparts. The authors analyze the impact of activation functions and training hyperparameters on the performance of SNNs and propose a new AutoML approach, SAFS, to optimize the activation functions and training hyperparameters of SNNs. The results demonstrate significant improvements in the accuracy of VGG-16, ResNet-18, and EfficientNet-B0 networks trained on CIFAR-10 and ImageNet-16 datasets, especially at high levels of sparsity.

**Overall Review:**

Positive:
1) Novel approach: The paper proposes a novel approach that combines hyperparameter optimization (HPO) and activation function search for sparse neural networks (SNNs). This approach is a significant contribution as there is no efficient approach for optimizing activation functions in SNN training.

2) Significant improvement in accuracy: The paper shows that the SAFS approach results in up to 8.88% and 6.33% absolute improvement in the accuracy of VGG-16 and ResNet-18 models, respectively, over the default training protocols, especially at high pruning ratios. This improvement is substantial and demonstrates the effectiveness of the proposed approach.

3) Addresses underexplored factors: The paper identifies and addresses two underexplored factors that impact the accuracy of SNNs, namely the choice of activation functions and training hyperparameters. By analyzing these factors and proposing the SAFS approach, the paper makes a significant contribution to the field of SNNs and deep learning.

Negative:
1) Limited comparison: This paper only compares its proposed approach, Sparse Activation Function Search (SAFS), with the default training protocols and the magnitude pruning method in 2015 for three popular DNN models trained on two datasets. More comparisons with other state-of-the-art methods and datasets would provide a more comprehensive evaluation of SAFS.

2) Lack of thorough analysis: The paper briefly mentions the impact of activation functions and training hyperparameters on the performance of sparse CNN architectures, but it does not provide a thorough analysis of these factors. A more detailed investigation of these factors could provide insights into why SAFS is effective and could lead to further improvements.

**Potential Impact On The Field Of Automl:**

This paper proposed a novel AutoML approach called Sparse Activation Function Search (SAFS) for learning novel activation functions for Sparse Neural Networks (SNNs) and combining them with a separate hyperparameter optimization regime. The authors demonstrate that the combination of these two approaches leads to significant performance gains, with up to 8.88% and 6.33% absolute improvement in accuracy for VGG-16 and ResNet-18, respectively, over default training protocols, especially at high pruning ratios.

**Review Confidence:**

3: You are fairly confident in your assessment. It is possible that you did not understand some parts of the submission or that you are unfamiliar with some pieces of related work.

**Review Rating:**

6: Borderline Leaning Accept: Technically sound paper where reasons to accept outweigh reasons to reject. Please use sparingly.

**Review Summary:**

The idea in this paper is novel, but empirical evaluation needs to be improved, including more experiments and more implementation details.

**Technical Quality And Correctness:**

This paper is technically sound in general, but I still have some concerns as follows:
1) Limited comparison: This paper only compares its proposed approach, Sparse Activation Function Search (SAFS), with the default training protocols and the magnitude pruning method in 2015 for three popular DNN models trained on two datasets. More comparisons with other state-of-the-art methods and datasets would provide a more comprehensive evaluation of SAFS.

2) Lack of thorough analysis: The paper briefly mentions the impact of activation functions and training hyperparameters on the performance of sparse CNN architectures, but it does not provide a thorough analysis of these factors. A more detailed investigation of these factors could provide insights into why SAFS is effective and could lead to further improvements.

---

> ### Author Response · Authors · 2023-04-28
> **Response to Reviewer YVAr**
>
> We thank you for your constructive feedback and your overall positive evaluation.
>
>
> ## Limited Comparison
>
> To provide further evidence, we have extended our experiments by optimizing the sparse LeNet-5 architecture with a 99% pruning ratio trained on the MNIST dataset. Our results show that SAFS improves the absolute classification accuracy of LeNet-5 by 15.53% (Table 2).
>
> ## Thorough Analysis
>
> [Ref3] showed that ReLU might not be the optimal activation function for any sparse neural network as ReLU reduces the network gradient flow. In a similar vein, the main reason for SAFS’s efficiency is the improvement in the network’s gradient flow by learning superior activation functions and corresponding hyper-parameters. We are going to add new results wrt gradient flow to the rebuttal submission of the paper.
>
> * [Ref3] Curci, Selima, Decebal Constantin Mocanu, and Mykola Pechenizkiyi. "Truly sparse neural networks at scale." arXiv preprint arXiv:2102.01732 (2021).

---

### Author Response · Authors · 2023-04-28
**General Response**

# General Response

We thank the reviewers for their helpful feedback. We are happy to see that the reviewers consider our approach to be novel (Reviewer YVAr) and adding a new perspective (Reviewer AwGP). We acknowledge the potential impact of applying AutoML to resource-constrained paradigms, which also happens to be our primary motivation to work on sparsity and be recognized for it (Reviewer JFfa). While we agree that this paper tackles the niche of sparsity (Reviewers JFfa, ziJv, uT44), we would like to extend on Reviewer ziJv’s c comment that this is a niche that is receiving an increasing amount of attention, given the recent applications it has seen, not just to image recognition, but also in sequential decision making problems and reinforcement learning scenarios  [Ref1].

* [Ref1] Liu, Vincent, et al. "The utility of sparse representations for control in reinforcement learning." Proceedings of the AAAI Conference on Artificial Intelligence. Vol. 33. No. 01. 2019.

We would also like to reiterate that the reason we tackle sparsity is specifically that the resource-constrained nature of the problem requires different ways of mitigating it. As we point out in Section 5.3, the same HPs for dense networks do not translate to sparse regimes. Additionally, given the potential applications of sparse networks to resource-constrained use cases, extracting as much performance as we can from these networks becomes very important. Hence, our motivation to perform an activation search on top of HPO. Through this work, in a similar vein to hardware-aware NAS, we aim to bring the attention of the AutoML Community to the resource-constrained regime, an area where we believe AutoML shines. To this end, we show that even standard techniques such as LAHC and SMAC3, combined with the added touch of SAFS can help us contribute to these applications. We aim to push further in this direction with follow-up work.

We would like to address general comments here, and address individual points as separate replies.


## Additional Experiments

In line with the reviewers’ requests for additional ablations and experiments, we have run and added the following to the paper:

1. Ablations
* Table 2: EfficientNet-B0 trained on ImageNet-16 with Swish activation functions and only HPO (only second stage)
2. Cost comparison
* Empirical costs of SAFS (Appendix B).
3. Additional results
* Table 2: Applying SAFS on the sparse LeNet-5 model trained on the MNIST dataset with a 99% pruning ratio.